# Multi-Omics Approach to Improved Diagnosis and Treatment of Atopic Dermatitis and Psoriasis

**DOI:** 10.3390/ijms25021042

**Published:** 2024-01-15

**Authors:** Lluís Rusiñol, Lluís Puig

**Affiliations:** 1Department of Dermatology, Hospital de la Santa Creu i Sant Pau, 08041 Barcelona, Spain; lrusinolb@gmail.com; 2Institut de Recerca Sant Pau (IR SANT PAU), 08041 Barcelona, Spain; 3Unitat Docent Hospital Universitari Sant Pau, Universitat Autònoma de Barcelona, 08025 Barcelona, Spain

**Keywords:** multi-omics, psoriasis, atopic dermatitis, immune mediated inflammatory diseases, genomics, epigenomics, proteomics

## Abstract

Psoriasis and atopic dermatitis fall within the category of cutaneous immune-mediated inflammatory diseases (IMIDs). The prevalence of IMIDs is increasing in industrialized societies, influenced by both environmental changes and a genetic predisposition. However, the exact immune factors driving these chronic, progressive diseases are not fully understood. By using multi-omics techniques in cutaneous IMIDs, it is expected to advance the understanding of skin biology, uncover the underlying mechanisms of skin conditions, and potentially devise precise and personalized approaches to diagnosis and treatment. We provide a narrative review of the current knowledge in genomics, epigenomics, and proteomics of atopic dermatitis and psoriasis. A literature search was performed for articles published until 30 November 2023. Although there is still much to uncover, recent evidence has already provided valuable insights, such as proteomic profiles that permit differentiating psoriasis from mycosis fungoides and β-defensin 2 correlation to PASI and its drop due to secukinumab first injection, among others.

## 1. Introduction

Psoriasis and atopic dermatitis are cutaneous immune-mediated inflammatory diseases (IMIDs) and, as such, belong to a spectrum of inflammatory conditions with two pathophysiologic poles: autoinflammatory and autoimmune. Autoinflammatory diseases are caused by the activation of the innate immune system, which leads to systemic inflammation [1,2]. On the other hand, autoimmune diseases are caused by the activation of the adaptive immune system, with high levels of autoantibodies and self-reactive lymphocytes, leading to inflammation and derangement of local tissues. Some diseases can be classified as either (predominantly) autoimmune (e.g., pemphigus vulgaris) or autoinflammatory diseases (e.g., familial Mediterranean fever), but psoriasis and atopic dermatitis are considered to be mixed-pattern diseases [2,3].

The incidence of IMIDs is growing in industrial societies; environmental changes combined with a genetic background trigger the development of IMIDs [4,5,6]. Recent evidence underpins the presence of shared common pathways among IMIDs [2,6,7], but knowledge of the immune factors that drive these chronic progressive diseases is still incomplete. Treatments of IMIDs are effective in providing clinical benefits to patients, but long-term disease control is still largely unmet [6].

Multi-omics, in the context of dermatology, refers to the use of various omics technologies to comprehensively study and analyze the skin and its related conditions. “Omics” fields encompass various biological data types, and in dermatology, this may include genomics (study of genes and DNA), transcriptomics (study of gene expression), proteomics (study of proteins), metabolomics (study of small molecules and metabolites), and microbiomics (study of the skin microbiome) [8,9,10]. The goal is to advance our knowledge of skin biology, better understand the underlying mechanisms of skin conditions, and potentially develop more precise and personalized approaches to diagnosis and treatment. For example, it can help identify specific biomarkers associated with skin diseases or responses to therapies, ultimately leading to more effective and tailored treatments for dermatological conditions. Due to the considerable length of the manuscript, we found it necessary to constrain the scope of our review. Hereby, we review the current knowledge of genomics, epigenomics, and proteomics in psoriasis and atopic dermatitis. Nevertheless, intriguing facts are also being unveiled by other omics technologies. For example, the combination of proteomics, single-cell transcriptomics, and spatial transcriptomics by Mitamura et al. uncovered the cellular crosstalk between the immune cells involved in skin lesions of atopic dermatitis [11]. Also, metabolomics has enabled the identification of urine metabolites associated with IMIDs, which could be valuable for their diagnosis [12].

## 2. Material and Methods

An electronic literature search was conducted on the Medline/PubMed database until November 2023, using Medical Subject Headings terms and pertinent medical terminology. The search criteria encompassed the following terms: ‘psoriasis’, ‘atopic dermatitis’, ‘genetic testing’, ‘genomics’, ‘genetics’, ‘epigenomics’, ‘epigenetics’, and ‘proteomics’. We considered original genome studies, reviews, systematic reviews, and meta-analyses that were specifically relevant to the genetic, epigenetic, or proteomic investigation of atopic dermatitis and psoriasis. Manuscripts written in the English language were eligible for inclusion, while letters to the editor, editorials, expert opinions, conference proceedings, studies exclusively involving patients with psoriatic arthritis, those focused on specific psoriasis treatments, or those centered on transcriptomics or proteomics analysis were excluded. The process of selecting publications was performed by two independent researchers (LR and LP), and any disparities were resolved through consensus.

## 3. Current Knowledge in Pathogenesis of Atopic Dermatitis and Psoriasis

Atopic dermatitis (AD) is the most common chronic cutaneous IMID, with a prevalence of 10% among adults and 20% in children [13]. The underlying pathophysiological mechanisms in AD are complex, encompassing a pronounced genetic susceptibility, epidermal dysfunction, and inflammation driven by T-cells [14,15,16,17,18]. Both the innate and adaptive immune systems contribute to its etiopathogenesis [14,16]. AD was initially considered a purely Th2-mediated inflammatory disease since most patients show high counts of eosinophils and high levels of immunoglobulin E (IgE). However, the immunological pathway of AD is complex, with predominant activation of Th2/Th22 and variable activation of Th17/Th1 lymphocytes; a biphasic switch from Th2 to Th1 responses has been reported in both acute and chronic cutaneous lesions of patients with AD [14,15,16]. The main cytokines related to AD etiopathogenesis are IL-4 and IL-13 [19,20]. They play a crucial role in the differentiation of Th2 cells and the production of IgE. Acute AD skin shows increased expression of Th2 cytokines, like IL-4, IL-5, and IL-13 [20]. In addition, IL-4 and IL-13 cause a disruption in epidermal barrier integrity by decreasing the number of main terminal differentiation proteins, such as filaggrin, loricrin, and involucrin. Levels of IL-13 correlate with the disease severity of AD [21,22,23].

Psoriasis is a chronic skin IMID with a prevalence of 1–3% in Western populations. The pathogenesis of psoriasis is characterized by an interaction of keratinocytes with the innate and adaptive immune systems [24,25,26]. Multiple theories have been put forth attempting to explain the pathogenesis of psoriasis [26,27,28]. The most accepted contemplates an initiation phase followed by a chronic inflammatory phase that is sustained by a feed-forward mechanism based on cytokine-mediated keratinocyte activation and proliferation [29,30]. Genetically predisposed patients become exposed to a trigger (trauma, infection, drugs) that will induce keratinocytes to undergo apoptosis or necrosis, releasing nucleic acids (DNA or RNA) that may induce a type I interferon-mediated autoinflammatory response, and potential autoantigens such as cathelicidin (LL37), ADAMTSL-5, and phospholipase A2 (PLA2G4D). LL37 interacts and forms complexes with self-DNA or -RNA that will trigger the innate immune system via Toll-like receptors (TLR) [31,32]. The activation of TLR7 and TLR8 leads to the production of interferon (IFN)-α and IFN-β by keratinocytes and plasmacytoid dendritic cells (pDCs), and interleukin (IL)-6 and tumor necrosis factor (TNF) by myeloid dendritic cells (mDCs). IL-6 induces differentiation of CD4+ naive T-cells into T helper (Th)-17 cells [33,34], whereas type I IFNs and TNF promote the secretion of IL-12 and IL-23 by mDCs [35,36]. IL-12 and IL-23 are clue cytokines in the immune chain reaction causing psoriasis. CD4+ naïve T-cells differentiate into Th1 cells upon exposure to IL-12, along with tumor growth factor (TGF)-β and IL-6 [37,38,39,40,41]. IL-23 leads to Th17 cell development and activates αβT-cells [37,38,39,40,41]. Th1-activated cells exhibit a distinctive cytokine secretion profile, including IFN-γ and TNF. Meanwhile, Th17-activated cells release IL-17, IL-22, and TNF [42]. Together, these cytokines induce keratinocyte proliferation, differentiation, and inflammatory activation, although IL-17A constitutes the main effector cytokine driving psoriasis pathogenesis and is essential for the development and maintenance of psoriasis plaques [42]. Inflammatory cytokines such as IL-1, IL-6, and TNF, chemokines (including CXCL1, CXCL2, and CXCL3), and AMPs (including S100A7/8/9, human-defensin 2, and LL-37) are produced by stimulated keratinocytes, which attract and activate immune cells and exacerbate psoriatic inflammation [40,43,44]. S100A8/A9 is overexpressed in keratinocytes and innate immune cells, and their transcripts are significantly overexpressed in psoriasis lesions compared to non-lesional psoriasis or atopic dermatitis (AD) skin [45]. Furthermore, psoriasis treatment has been shown to reduce S100A8/A9 levels. Christmann et al. identified the induction of S100-alarmins in an imiquimod-induced murine model of psoriasis-like skin inflammation, which was associated with increased expression of IL-1*α*, IL-6, IL-17A, or TNF [46]. However, recent evidence has observed that lower epidermal levels of S100A9 in mice lead to more severe psoriasis skin lesions [47].

The maintenance phase of psoriatic inflammation is driven by this positive feedback loop between keratinocytes and T lymphocytes, enhancing further epidermal hyperplasia and a sustained inflammatory response.

Psoriasis encompasses multiple clinical variants, such as pustular psoriasis, guttate psoriasis, and nail psoriasis, among others. Guttate psoriasis was previously thought to be closer to contact dermatitis than psoriasis; however, novel insights by using gene expression profiling and gene set enrichment scores have been observed that are more similar to chronic psoriasis [48]. Regarding pustular psoriasis, in contrast to psoriasis vulgaris, where the IL-23/17 axis plays a pivotal role, pustular psoriasis shows hyperactivation of innate immunity, prominently involving the IL-36 axis [49]. Analysis of gene expression in skin biopsy specimens obtained from individuals affected with either generalized pustular psoriasis (GPP) or plaque psoriasis has unveiled discernible patterns. Specifically, GPP lesions manifest increased expression of IL-1 and IL-36, coupled with diminished levels of IL-17A and interferon-c, when juxtaposed with lesions characteristic of plaque psoriasis [49]. The pathomechanisms of ungual psoriasis remain elusive; however, a variant in IL1RN has been identified in patients with nail psoriasis. IL1RN functions to regulate the proinflammatory activity of IL-1A. The latter has been demonstrated to induce nail changes, suggesting a potential association with nail involvement in patients affected by psoriasis [50,51].

The clinical manifestations and course of both psoriasis and AD are highly variable; most patients achieve satisfactory responses with currently available treatments, but clinical and therapeutic challenges can be vexing in some cases.

## 4. Omics of Atopic Dermatitis and Psoriasis

### 4.1. Genomics

#### 4.1.1. Atopic Dermatitis

AD has a strong genetic background, with 70% to 80% heritability [5,52]. The development of AD is significantly influenced by genetic factors, and a family history of AD is the strongest risk factor for this disease [13].

Genetic linkage studies, genome-wide association studies (GWAS), candidate gene techniques, and, more recently, next-generation sequencing (NGS) technologies have all been used to identify the genetic risk factors underlying AD and psoriasis. Genetic linkage studies correlate the inheritance of genetic markers with the existence of clinical features. The first genome-wide linkage study for AD in 2000 identified a major locus of susceptibility on 3q21 [53]. Additional genome-wide linkage studies on AD were performed in the following years, leading to the identification of numerous other loci, including 1q21, 3p, 17q, 18q, and 11.13q. But these loci were frequently excessively wide, necessitating labor-intensive fine mapping [54]. The most recent and largest genome-wide association meta-analysis of AD—including 65,107 AD cases and 1,021,287 controls—was published in October 2023. The authors identified a total of 91 associated loci [55]. Among them, 81 loci were initially identified in individuals of European ancestry, but further studies showed that most of them were replicated in data from other population ancestries. Eight loci were detected in at least one of the populations: European, Latino, and African ancestry, and the remaining 2 loci could be specific to individuals of East Asian ancestry (pending further studies). As stated, most loci identified were shared among ancestries of different populations. However, some loci shared between European, Latino, and Japanese ancestries did not show evidence of association in individuals of African ancestry. Given the distinct AD phenotypes identified among European, Asian, and African individuals, it is tempting to hypothesize that variations in genetic associations at certain loci may be a factor accounting for these clinical phenotypes [55].

The candidate gene technique used to uncover loci linked to the onset and severity of AD has been employed in more than 30 studies. In this method, variation in genes suspected of being involved in disease pathogenesis is compared between affected and healthy individuals [56].

Progress in research has led to the identification of more than 70 genes associated with AD; they can be classified into five groups: genes leading to cutaneous barrier dysfunction, genes associated with altered innate responses, genes associated with acquired immune responses, genes associated with stress responses of the keratinocytes, and genes involved in the metabolism of vitamin D [57,58], as detailed in Table 1.

One of the most well-known genes linked to both disease susceptibility and disease severity is *FLG*, the gene encoding for filaggrin, the main protein responsible for the maintenance of cutaneous epithelium integrity [58]. Null mutations in the *FLG* gene result in a deficiency in the epidermal barrier [59,70,71,72]. Patients with heterozygous mutations have an 8-fold increased risk of acquiring AD, while carriers with two mutant alleles are almost always affected by it [58,73].

Apart from *FLG*, two other key genes have been identified: *OLOV1* (Ovo Like Transcriptional Repressor 1) and *IL-13*. The former is a transcriptional factor that regulates filaggrin expression [60]; the latter mediates Th2 cell responses [57,58], which predominate in AD. The expression of cutaneous barrier proteins such as filaggrin, loricrin, involucrin, cell adhesion proteins, desmosine, and claudins is decreased by Th2 responses with secretion of IL-4 and IL-13 [57,58]. Additionally, Th2 responses disrupt the homeostasis of other epithelia, which results in an unbalanced immune response that is ultimately linked to systemic inflammation, airway hyperresponsiveness, and food allergies [57,74,75,76]. *IL-4* gene polymorphisms have also been related to an increased susceptibility to AD [66]. Indeed, IL-4 and IL-13 are successfully targeted by novel biologics used in AD treatment [77].

IL-10 is an anti-inflammatory cytokine that contributes to the modulation of acquired immune and anti-inflammatory responses [67,78]. Polymorphisms in the promoter region of the *IL-10* gene can affect the function of IL-10. Among the several polymorphisms identified in the meta-analysis by Zhao et al. two stand out: IL-10-819G/A and IL-10-1082G/A. The former seems to be associated with an increased risk of AD in Caucasian populations, whereas the latter would be associated with an increased risk of AD in Asian populations [67]. However, according to another meta-analysis performed in the same year, available evidence did not support a strong genetic relationship between the IL-10-1082G/A polymorphism and AD risk [78].

IL-6 plays an important role in host defense mechanisms and can act as both a pro-inflammatory cytokine by stimulating the production of IL-4 and IL-5—predominant Th2 cytokines in acute AD—and as an anti-inflammatory cytokine by inhibiting IL-1 and TNF and activating IL-10 and IL1Rα. Single nucleotide polymorphisms (SNPs) in the *IL6* gene have been associated with an increased inflammatory response, as in AD [68]. In addition, SNPs in *IL6R* have also been associated with an increased risk of AD [69].

Alterations in genes codifying proteins that participate in the integrity of the epidermis have also been related to AD. For example, the *COL5A3* (Collagen type V Alpha 3 Chain) and the Matrix metalloproteinase *(MMP)-9* genes [61]. *COL5A3* codifies for type V collagen, which is found in the dermoepidermal junction and in the extracellular matrix associated with type I collagen. The role of MMP9 in tissue remodeling and collagen deposition is well known; it also participates in inflammation by facilitating cellular traffic, including neutrophils and eosinophils. MMP9 would mediate the degradation of type V collagen, leading to eczema susceptibility [61]. *COL29A1* gene, which encodes for collagen expressed in the skin, lung, and gastrointestinal tract, has also been associated with AD by genetic linkage. In the outer layers of the epidermis, a lack of collagen XXIX expression would derange keratinocyte cohesion and the integrity and function of the epidermis [62].

Other genes that have been implicated in the pathogenesis of AD are *SPINK5* (Serine Peptidase Inhibitor Kazal Type 5), which produces a serine protease inhibitor that contributes to maintaining the skin barrier [63,79], signal transducer and activator of transcription (*STAT*), thymic stromal lymphopoietin (*TLSP*), interferon regulatory factor 2 (*IRF2*), *TLR 2*, high-affinity IgE receptor (*FcεRI α*), vitamin D receptor (*VDR*) [57], *ACTL9* (Actin Like 9), *KIF3A* (Kinesin Family member 3A) [64], *LCE1D* (Late Cornified Envelope 1D), *SPRR3* (Small Proline Rich Protein 3), and *S100A3* (S100 Calcium Binding Protein A3) [65].

#### 4.1.2. Psoriasis

Psoriasis is a complex disease characterized by multifactorial inheritance with a strong genetic background and greater than 60% heritability [80,81]. Epidemiological studies revealed a higher prevalence of psoriasis vulgaris among first- and second-degree relatives of affected individuals compared to the general population [82]. Approximately 30% of individuals with psoriasis vulgaris have a first-degree relative affected by the condition. In cases where both parents and a sibling are affected, there is a 50% likelihood of another child developing psoriasis vulgaris. However, if the sibling is affected without the parents’ showing symptoms, the risk decreases to 8%. Moreover, the risk of psoriasis vulgaris is notably elevated in monozygotic twins, being two to three times greater than in dizygotic twins [82].

A minimum of 12 distinct loci suspected to contain genes related to psoriasis susceptibility (PSORS) were identified by genetic linkage studies [83]. However, most of these findings could not be replicated, underscoring the limitations of genetic linkage [80]. *PSORS1* was the first locus associated with psoriasis susceptibility; it is the locus with the greatest genetic effect and represents 35–50% of psoriasis heritability explained by known loci [80]. PSORS1 contains HLA-Cw6, the major psoriasis susceptibility allele, located at 6p21.3 [84], as well as the candidate gene corneodesmosin (*CDSN*), which codifies for a desmosomal protein involved in keratinocyte cohesion and desquamation [85]. *PSORS2* and *PSORS4* loci have also been validated. *PSORS2* contains the *CARD14* (caspase recruitment domain family member 14) gene, which encodes for a nuclear factor kB (NF-kB) activator; some of its variations have been linked to common and rare variants of psoriasis [86,87]. Lastly, *PSORS4* contains *LCE* genes, which codify stratum corneum proteins that mediate in terminal epidermal differentiation [88].

Genetic linkage studies have been replaced by newer and more potent analytical techniques that permit a thorough examination of genetic risk variation at the genome-wide level (GWAS, NGS). Genome-wide association studies (GWAS) employ highly optimized microarrays capable of efficiently and robustly genotyping millions of genetic markers throughout the genome [80]. By leveraging substantial sample sizes, GWAS facilitates the detection of even minor distinctions in allele frequencies between individuals with the disease and those without and provides a significantly more potent approach compared to linkage analysis. However, GWAS only permits uncovering statistical relationships; in order to identify potential causal susceptibility alleles, genotyping arrays with dense coverage in regions of interest have been used. To enhance the information obtained by GWAS, additional SNPs not included in the GWAS have been genotyped [80]. These studies allowed the identification of additional susceptibility loci and uncovered genes involved in innate immunity, such as *DDX58* and *CARD14* [89], and genes related to type I interferon signaling (*IFIH1* and *TYK2*) [90]. Nevertheless, nowadays the prevailing practice consists of conducting genome-wide imputation through freely available computational resources. Imputation plays a crucial role in enabling extensive meta-analyses in psoriasis by amalgamating data produced by diverse GWAS platforms [80]. In fact, the largest genetic study of psoriasis susceptibility undertaken to date, a GWAS meta-analysis of 18 international case-control studies, has identified genetic variants at 109 distinct loci, 45 of which had not been previously reported [91]. They include susceptibility variants located within loci that encode therapeutic targets such as IL17RA and AHR, along with deleterious coding variants that suggest potential novel drug targets, including those found in *STAP2*, *CPVL*, and *POU2F3* [91].

Genes that have been linked to psoriasis participate in pathways of the epidermal barrier, the innate immune system, and the adaptative immune system, as detailed in Table 2 and discussed as follows.

Structural and functional alterations of keratinocytes participate in the pathogenesis of psoriasis, especially as regards its onset [102,106]. A higher risk of psoriasis has been associated with alterations in *LCE* genes, especially *LCE3B* and *LCE3C*, genes coding for defensins (*DEFB4*), and the *GJB2* gene [92,93,94].

The innate immune system plays a critical role in psoriasis pathogenesis. Its stimulation is necessary for the subsequent activation of the adaptive immune system [83]. Gene candidates for innate immunity pathways in psoriasis have been found through numerous single nucleotide polymorphisms (SNPs) [83]. Transcription of these innate immune genes may reduce the threshold necessary to activate the pathogenic adaptive immune response [83,105]. NF-kB plays a critical role in mediating the transcription of several genes that participate in the innate response [107,108]. Some of them stand out: *C-REL* [95], which is also related to keratinocyte growth; TRAF3-interacting protein 2 (*TRAF3IP2*) [96], which mediates the interaction of the innate and adaptive immune systems; and genes encoding CARD proteins [86]. The interferon pathway and antiviral response genes, such as Interferon Induced with Helicase C Domain 1 (*IFIHI*) and RNA sensor RIG-I (known as *DDX58*), have also been associated with psoriasis [109]. Alterations in the downregulation of the NF-kB pathway, due to mutations in negative regulators, have also been related to psoriasis susceptibility. These mutations reduce the ability to control inflammation and are as important as mutations inducing overactive immune responses [110]. Some of the genes implicated are TNF-α-inducible protein 3 (*TNFAIP3*), TNFAIP3 interacting protein 1 (*TNIP1*), the NF-kB inhibitor α (*NFKBIA*), and the zinc finger DHHC-type containing 23 (*ZC3H12C*) [89,92,97].

The major histocompatibility complex (MHC) accounts for one-third of the total genetic influence on psoriasis, highlighting the important pathogenetic role of the antigen presentation pathway in psoriasis [102]. PSORS1 risk variant contains HLA-Cw06. This is present in 20–50% of the patients with psoriasis, compared to 16% of individuals without the disease. Psoriasis can be considered an MHC-1-opathy, in which self-immunogenic peptides are presented by the psoriasis-linked MHC-1 alleles to effector cells, leading to the inflammatory response [98].

Interactions between MHC class I risk alleles and variations in endoplasmic reticulum aminopeptidase 1 (*ERAP1*) have been identified by GWAS. ERAP1 is an aminopeptidase that prepares peptides for effective antigen presentation. These interactions have been linked to psoriasis risk [99]. Finally, other loci have also been associated with a higher risk of psoriasis [111]. For example, the MHC class I polypeptide-related sequence A (*MICA*), is found near the HLA-B locus in the MHC region. MICA acts as a ligand for an activating receptor in natural killer cells, NKT cells, and T-cells [83].

The Th1 signaling pathway carries out a crucial role in the pathogenesis of psoriasis, whose lesions have increased numbers of T-cells producing interferon (IFN)-γ [102,112]. IL-12 drives Th1 polarization of T-cells and secretion of IFN-γ, TNF, and IL-2. Multiple genetic psoriasis susceptibility loci affect different parts of the Th1 signaling pathway, including *IL-12B* (coding for the p40 subunit of IL-12) [100], tyrosine kinase 2 (*TYK2*) [101], *ZC3H12C* (transcript involved in macrophage activation) [89], *STAT5A*, *STAT5B*, and *ILF3* (interleukin enhancer-binding factor 3) [102].

The IL-23/IL-17 pathway plays a major pathogenetic role in psoriasis since Th17 cytokines are the main effectors of keratinocyte alterations and inflammation in this disease [112,113]. Multiple SNPs associated with psoriasis risk have been identified in genomic regions corresponding to both subunits of IL-23 (p19 and p40) and *IL23R* [100,114]. In addition, the activation of IL-23R leads to downstream signaling through multiple molecules such as TYK2, JAK2, and STAT3 [103,104]. Mutations in genes coding for these molecules have also been shown to be associated with psoriasis [103,104]. Other genetic variants related to IL-23 and psoriasis risk include suppressor of cytokine signaling 1 (*SOCS1*) and ETS Proto-Oncogen 1 (*ETS1*) [102]. On the other hand, no association has been yet established between the *IL17A* gene and psoriasis, but an SNP in the *IL17R* gene is related to disease susceptibility [115]. *IL22* has been associated with psoriasis risk quantitatively by GWAS: a higher number of *IL22* gene copies leads to a higher risk of nail psoriasis [103].

The balance between Th17 and regulatory T-cells (Treg) is also altered in psoriasis, where Treg develop expression of RORγt and become functionally impaired; this can be partially explained by polymorphisms in genes such as *TNF*, *IL12RB2*, and *IL12B* [116,117,118,119].

The pathogenesis and genetic background of plaque psoriasis differ from that of generalized pustular psoriasis, an autoinflammatory disorder with special implications for neutrophils and IL-36 [120]. Loss-of-function mutations in *IL36RN* (coding for the antagonist of the IL-36 receptor) result in DITRA (Deficiency of The Interleukin-36-Receptor Antagonist), a monogenic autoinflammatory disorder with earlier onset and more severe presentation of pustular psoriasis [106,120]. Activating mutations of *CARD14* (caspase recruitment domain family member 14), which codes for an inductor of NF-kB signaling expressed in keratinocytes, have also been observed in plaque psoriasis, psoriatic arthritis, and generalized (as well as localized) pustular psoriasis [120]. Other genes with mutations identified in patients with generalized pustular psoriasis include the adaptor-related protein complex 1 subunit sigma 3 (*AP1S3*) [121], mutations in Myeloperoxidase (*MPO*) genes [122], Serpin Family A Member 1 (*SERPINA1*) and *SERPINA3*, *TNIP1,* and *IL1RN* [120].

#### 4.1.3. Mendelian Randomization

Mendelian randomization (MR) is a statistical technique that utilizes genetic variants to evaluate the causal relationship between a specific exposure and the occurrence of a particular disease and has been increasingly used in the study of AD and psoriasis [13]. The increasing availability of GWAS has contributed to the growing prominence of MR in exploring potential causal associations within observational studies. Genetic variants are used as surrogates for the targeted risk factor, and MR mitigates the potential for confounding and reverse causation, thereby yielding a more reliable estimate of causality.

##### Mendelian Randomization in AD

Multiple studies facing MR in AD have been published lately, including a systematic review in which 30 MR studies were included [123]. This systematic review points body mass index, gut microbial flora, the IL-18 signaling pathway, and gastroesophageal reflux disease as causal factors for AD. On the other hand, AD would be causal for several medical conditions, including health failure, rheumatoid arthritis, and conjunctivitis. Prior studies also disclosed that AD would carry a greater risk of type 2 diabetes and alopecia areata [124,125], whereas no association between AD and higher cancer risk has been found [126].

##### Mendelian Randomization in Psoriasis

Extensive Mendelian randomization approaches have been performed to identify a genetic relationship between psoriasis and several comorbid diseases. A systematic review, in which 27 MR studies were included, was recently published [127]. Exposures like smoking, obesity, cardiovascular disease, and Crohn’s disease were identified as causal factors for psoriasis and psoriatic arthritis. Whereas, psoriasis was found to be causally associated with cardiovascular complications (myocardial infarction, large artery stroke), a mild risk for lung cancer, COVID-19, and Parkinson disease.

Chalitsios et al. aimed to determine different clusters of co-existing long-term conditions (LTC) in patients with psoriasis [128]. By employing latent class analysis, the cross-sectional data of individuals diagnosed with psoriasis from the UK Biobank underwent examination to delineate specific co-morbidity profiles associated with psoriasis. Linkage disequilibrium score regression was subsequently used to calculate the genetic correlation existing between psoriasis and LTC. They identified 5 different clusters of psoriatic patients [128]. Patients included in clusters 1, 2 and 4 were associated with metabolic syndrome traits, cardiovascular diseases, airway and pulmonary diseases, and depression. Psoriatic patients included in cluster 3 were more prompt to suffer from psoriatic arthritis and rheumatoid arthritis, along with depression and hypertension. Finally, patients in cluster 5 were “relatively healthy”, showing a low probability of developing multimorbidity.

#### 4.1.4. Ferroptosis, Pyroptosis, and Cuproptosis

Cell death plays a critical role in embryonic development, cell fate determination, and the maintenance of immune homeostasis. It is categorized into necroptosis, apoptosis, pyroptosis, cuproptosis, ferroptosis, and necrosis. In this chapter, we will focus on cuproptosis, ferroptosis, and pyroptosis [129].

Ferroptosis, a form of iron-dependent cell death, results in a toxic accumulation of reactive oxygen species. The disruption of iron homeostasis and the oxidation of phospholipids can instigate the occurrence of ferroptosis. Due to its distinctive mechanism, ferroptosis may play a role in determining cellular outcomes, the advancement of inflammatory processes, and various pathological conditions [130].

Pyroptosis, a recently identified form of programmed cell death, is orchestrated by pyroptotic caspases [129]. This modality is characterized by the prompt rupture of the plasma membrane, leading to the release of inflammatory intracellular contents. Recognition of pathogen-associated molecular patterns (PAMPs), damage-associated molecular patterns (DAMPs), and lipopolysaccharide (LPS) by specific inflammasomes and caspases induces the activation of pyroptosis pathways. There are two pyroptosis pathways: canonical and non-canonical. The former is caused by PAMPs and DAMPs, which permit the formation of inflammasomes. Inflammasomes cleave procaspase-1 into caspase-1; activated capsase-1 cleaves gasdermin D (GSDMD) and the maturation of IL-1B and IL-18. Whereas the non-canonical pathway, lipopolysaccharide can activate caspase-4/5/11 to induce pyroptosis by cleavage of GSDMD. Increasing evidence suggests the profound involvement of pyroptosis in infectious diseases, hematologic disorders, and tumorigenesis. Furthermore, activated inflammasomes triggered by both PAMPs and DAMPs are implicated in the initiation of chronic inflammation and autoimmune diseases.

Cuproptosis represents a novel type of cellular death associated with mitochondrial metabolism and facilitated through protein lipoylation [131]. The occurrence of cell death induced by copper ionophores depends predominantly upon the intracellular accumulation of copper. Research findings indicate that FDX1 and protein fatty acylation play pivotal roles as regulators in the context of cuproptosis.

##### Ferroptosis, Pyroptosis, and Cuproptosis in AD

Pyroptosis has been related to AD pathogenesis [132]. Research has focused on the identification of pyroptosis-related genes (PRGs). Recent studies have identified an increased expression of inflammasome PRGs, such as NLR Family Pyrin Domain Containing 3 (NLRP3) and Absent In Melanoma-2 (AIM2) in AD, which have been related to epidermal inflammation [133,134]. Also, genes of gasdermin C and GSDMD were found to be highly expressed in lesional skin-derived keratinocytes [135,136]. TNF, which plays a role in the pathogenesis of allergic inflammation in AD, forms a complex with caspase-8 and GSDMC. Pyroptosis mediated by the complex TNF-Caspase8-GSDMC could trigger keratinocyte inflammation and death [136]. All these findings suggest that PRGs may play a pivotal role in AD pathogenesis, and ultimately, the identification of pyroptosis-related biological markers (PRBMs) could provide a novel perspective on the diagnosis and treatment of AD [129].

To our knowledge, no ferroptosis or cuproptosis alterations related to AD pathogenesis have been identified yet.

##### Ferroptisis, Pyroptosis and Cuproptosis in Psoriasis

Ferroptosis is an iron-dependent, lipid peroxidation-mediated cell death pattern that is intricately linked to inflammation within psoriatic lesions [137]. The extent of the impact of ferroptosis on psoriasis remains largely undetermined, but keratinocytes isolated from psoriatic lesions exhibit irregularities in lipid metabolism and expression. At the individual cell level, there is pronounced activation of lipid oxidation and peroxidation in psoriasis lesional keratinocytes [138].

Ferroptosis can be activated by the generation of reactive oxygen species (ROS) [137]. ROS induces GPX4 (glutathione peroxidase 4) deletion. In normal conditions, GPX4 inhibits ferroptosis [139]. The GPX4 deletion decreases keratinocyte adhesion and increases intracellular lipid peroxidation [140]. GPX4 expression is decreased in psoriatic skin lesions compared to healthy skin. Furthermore, augmentation in the cellular import of iron indicates the activation of ferroptosis in psoriasis lesions [141]. Other genes that are related to ferroptosis and regulate the immune microenvironment in psoriasis are PEBP1, PRKAA2, and ACSF2 [138]. PEBP1 mediates ferroptosis vulnerability due to GPX4 deletion, and PRKAA2 induces ferroptosis by inhibiting the transcription of SLC7A11 [142]. Finally, a positive correlation between lipid oxidation and the Th22/Th17 pathway at a single-cell level has been demonstrated: Fer-1 (a specific inhibitor of ferroptosis) decreases inflammation in mice with imiquimod-induced psoriasis, with significantly decreased expression of TNF, IL-6, IL-1*α*, IL-1*β*, IL-17, IL-22, and IL-23 [139].

Pyroptosis plays a crucial role in the pathogenesis of disorders with an aberrant Th17 immune response, like psoriasis [143]. Keratinocytes apparently engage cell death by pyroptosis, paradoxically accelerating cell proliferation in the pathogenic environment of psoriasis. In addition, GSDMD-mediated pyroptosis would contribute to the inflammatory environment by allowing the release of DAMPs and inflammatory cytokines. Effective inhibition of pyroptosis with topical application of disulfiram 2% and 5% has been reported to reduce psoriasis-like inflammation in a murine model [143].

Cuproptosis and elevated copper levels in serum may be involved in the pathogenesis of psoriasis by mechanisms largely unknown [131]. The accumulation of intracellular copper leads to cell death, and the cuproptosis-related genes MTF1, ATP7B, and SLC31A1 are increasingly expressed in patients with psoriasis compared to patients without psoriasis [131].

### 4.2. Epigenomics

Epigenetics consists of the examination of reversible and inheritable alterations in gene expression that occur without changes in the DNA sequence [144]. Epigenetic modification of gene expression occurs either at the transcriptional or post-transcriptional level. The former is mediated by modifications in the chromatin structure that lead to the activation of transcription factors for specific genes without impacting the underlying DNA sequence (Figure 1). Epigenetic mechanisms at the post-transcriptional level are driven by non-coding RNAs (ncRNAs) that alter the structural properties of RNA, DNA, and proteins, modifying gene expression.

Epigenetic mechanisms are sensible to external stimuli, representing a link between genetics and the environment. Therefore, epigenetic mechanisms may have the potential to modify the susceptibility to diseases where environmental factors contribute to their etiopathogenesis, such as AD and psoriasis. Nonetheless, the impact of epigenetics is most important during embryogenesis, when pluripotency cells are highly common. The stabilized epigenome in differentiated cells is less susceptible to modification by environmental factors [145]. Epigenetic changes in patients with AD and psoriasis involve genes that regulate immune response and inflammatory processes and those that code for structural proteins of the epidermis [56,146,147,148,149].

The epigenome characteristics associated with any disease can be described according to the epigenetic mechanisms involved, which can alter gene transcription (namely, DNA methylation and histone modifications) or gene translation via non-coding RNAs, including microRNA (miRNA), long non-coding RNA (lncRNA), and circular RNA (circRNA) [56,144,150].

#### 4.2.1. DNA Methylation

DNA methylation is among the most prevalent epigenetic mechanisms that regulate gene expression. This process targets specific DNA sections known as CpG sites, where cytosine is followed by guanine. Around 70% of the human genome’s proximal promoter regions are made up of these main sequences [58,150,151]. DNA methylation consists of adding a methyl group to the C5 position of CpG dinucleotides (cytosine becomes 5-methylcytosine) by DNA methyltransferases (DNMTs); this allows methylcytosine-binding proteins (MBPs) to attach and induce chromatin compaction [152]. When DNA methylation occurs in the promoter region of a gene, this leads to suppression of gene expression since transcriptional factors cannot bind (Figure 1). Conversely, loss of DNA methylation permits gene expression due to chromatin decompaction [152]. Cytosine demethylation is catalyzed by ten-eleven translocation (TET) enzymes (Figure 2).

Several alterations in DNA methylation have been identified in AD and psoriasis, with potential etiopathogenetic implications.

##### DNA Methylation in AD

The DNA methylation profile observed in the skin of patients with AD is different from that of healthy individuals [56]. The keratinocytes obtained from AD lesions showed demethylation of the thymic stromal lymphopoietin gene (*TSLP*) promoter, which led to overexpression of the corresponding Th2-biasing alarmin in the epidermis [153]. The promoter of *FCER1G*, which codes for the high-affinity IgE receptor FcεRI, was also shown to be demethylated in the monocytes of AD patients, leading to the overexpression of this receptor on the surface of these cells [154].

AD pathogenesis in many patients is primarily driven by mutations that result in decreased filaggrin expression. According to research by Ziyab et al., heterozygotic carriers of the null mutations R501X, 2282del4, and S3247X may exhibit excessive methylation of the filaggrin gene (*FLG*). This interacts with epigenetic regulation to increase the risk of AD in the carriers [155]. Variants of another gene implicated in skin barrier function, kinesin family member 3A (*KIF3A)*, are associated with an increased risk of AD and present alterations in DNA methylation [156,157].

Epigenetic changes play a fundamental role in the differentiation process of subpopulations of T lymphocytes [158,159]. Activated Th2 lymphocytes produce IL-4, IL-5, and IL-13 cytokines. Their epigenetic activation is related to the demethylation of *IL-13* and *IL-4* gene promoters and the methylation of the *IFG* gene promoter. On the other hand, activation of Th1 lymphocytes can be caused by methylation of CpG islands of the *IL-4*, *IL-13*, and *IL17A* gene promoters, methylation of the *RORC* gene transcription factor, and demethylation of *IFG* and *TBX21* [158,159,160]. Treg deficiency, which is one of the hallmarks of AD pathogenesis, is associated with methylation of the *FOXP3* promoter and demethylation of *RORC* [161,162,163].

##### DNA Methylation in Psoriasis

DNA methylation in psoriasis occurs in immune-active sites other than the skin. Moreover, this is associated with disease severity, the distribution of lesions, and the different tissue types involved [164]. Epigenome-wide methylation studies have shown differentially methylated genes and pathways in psoriatic lesions compared to skin from healthy controls or even to non-lesional skin of patients with psoriasis [165,166,167], even though the methylation pattern of the latter is closer to that of skin in healthy individuals than to psoriatic skin lesions [164]. In recent studies comparing psoriasis-involved skin to adjacent non-lesional skin, differences in CpG methylation were found at the promoters of known PSORS genes, such as *S100A9*, *PTPN22*, *SELENBP1*, *CARD14*, and *KAZN* [167]. In addition, DNA methylation differences can also be related to the histological features of psoriasis lesions. Patients with Munro’s microabscess present a characteristic DNA methylation profile with the involvement of genes related to the chemotaxis of neutrophils [167]. *S100A8* (involved in epidermal differentiation), *CYP2S1* (metabolism of retinoic acid), and *EIF2C2* (with a role in RNA processing) are also differentially methylated in psoriatic lesions compared with non-lesional skin and with the normal skin of healthy controls [165]. The *MGRN1* gene codes for E3 ubiquitin-protein ligase, which is involved in the degradation of misfolded proteins. Its methylation pattern is different between psoriatic scales and psoriatic skin lesions [168].

Tissue-resident memory T-cells in the skin have been related to the local relapse of psoriasis. DNA methylation alterations could be involved in this local memory since they do not revert completely to lesional skin after treatment. Indeed, distinct methylation patterns have been observed between never-lesional skin and resolved psoriasis lesions [169]. Methylation differences in Wnt and cadherin pathway genes have also been shown between never-lesional psoriatic skin and healthy skin in volunteers, suggesting that uninvolved skin could be a pre-psoriatic condition with an underlying vulnerability to disease [170].

Variations in DNA methylation have been found in peripheral blood mononuclear cells (PBMCs) from patients with psoriasis. The expression of methyl-CpG binding domain protein 2 (MBD2) and methyl-CpG binding protein 2 (MeCP2), two crucial regulators of DNA methylation, is markedly downregulated in PBMCs from psoriatic patients, whereas the methyltransferase DNMT1 is overexpressed [171]. Likewise, mesenchymal stem cells from psoriasis patients have been found to exhibit differential methylation of the promoters of genes involved in cell interaction, signal transduction, nucleotide degradation, skin differentiation, and cell motility [172].

#### 4.2.2. Histone Modification

Histones are highly conserved proteins that are found in cell nuclei as structural components of chromatin and are involved in the control of gene expression. The accessibility of DNA sequences can be altered by post-transcriptional changes of histones, hence modifying the transcription of these DNA sequences [167]. Histone acetyltransferases (HATs) and histone deacetylases (HDACs) are the enzymes that maintain the general equilibrium between histone acetylation and deacetylation.

A wide spectrum of histone modifications with potential pathogenic implications have been described in psoriatic and AD skin. The mechanisms that influence chromatin compression due to post-translational modifications of histones are acetylation, phosphorylation, methylation, ubiquitination, and SUMOylation (Figure 3) [58].

While histone deacetylation loosens the chromatin structure and activates gene transcription, acetylation causes chromatin to become more tightly packed and prevents transcription (Figure 1). On the other hand, methylation of histones will cause activation or suppression of the transcription, depending on the site of methylation and the number of methyl groups introduced.

##### Histone Modification in AD

Histone alterations related to AD pathogenesis can involve processes related to epidermal dysfunction, epidermal differentiation, and immune dysfunction. Regarding epidermal dysfunction, trimethylation of histone H3 at its Lys 27 residue by the histone transmethylase EZH2 prevents premature recruitment of activator protein 1 (AP1) to activate transcription of several structural genes required for epidermal differentiation [173,174]. Also, mixed-lineage leukemia (MLL), along with the multisubunit (WDR5, RbBP5, ASH2L, and DPY30) complex, catalyzes the trimethylation of H3K4, regulating the expression of genes related to differentiation in human skin [175]. Finally, histone hyperacetylation promoted by 5-azacytidine (5AC) and sodium butyrate (NaB) leads to keratinocyte differentiation through increased expression of Sprr1/2 and involucrin in human keratinocytes [176].

Decreased expression of proteins involved in tight junctions, such as transmembrane proteins and intracellular cytoplasmic proteins, has been observed in AD. Steelant et al. identified a new pathway associated with histone deacetylase (HDAC) activation and epidermal barrier dysfunction. In addition, they demonstrated that the application of HDAC inhibitors restored barrier integrity and reduced hyperresponsiveness in asthmatic mice [177].

Regarding immune dysfunction and histone modifications, Harb et al. demonstrated that placental histone hyperacetylation was linked to a low incidence of food allergen sensitization in children [178]. They noticed H3 acetylation of the *SH2B3* gene, H3 acetylation of the *IFNG* gene, and H4 acetylation of the *HDAC4* gene. The latter boosted HDAC4 expression in the placenta. HDAC4 possesses deacetylase activity for cytoplasmic nonhistone proteins, like STAT1. According to the authors, the higher placental expression of HDAC4 would lead to genome-wide deacetylation of immune genes and/or transcription factors involved in Th1 tilting. Thus, would enhance early immune polarization, aiding newborns to avoid IgE sensitization [178]. There is abundant additional evidence that histone modification and DNA methylation participate in the differentiation process of T lymphocytes [158,159,160].

##### Histone Modification in Psoriasis

Several histone modifications have been described with potential implications for psoriasis pathogenesis. For example, reduced levels of methylated H3K27 have been correlated with Th17 differentiation [179]. Other alterations reported are methylation at cystine 9 of histone 3 (H3K9), which can modify IL23 expression in keratinocytes, and H3K4 methylation in PBMCs from psoriasis patients [180,181]. H3K27 and H3K4 methylation of cystines 4 and 27 of histone 3 could influence the response to treatment since their levels have been found to differ between responders and non-responders to biological treatment [181].

Dysregulation of HATs and HDACs has been found in lesional psoriasis skin compared to healthy skin [167]. Elevated levels of HDAC-1 in psoriatic skin might lead to VEGF overexpression, endothelial cell proliferation, and keratinocyte survival [182]. High levels of H3K9 and H3K27 acetylation have been found in the IL17A promoter region in the immune cells of patients with psoriasis; they would promote Th17 differentiation and psoriasis development [183].

Acetylated histones are recognized by BET proteins that mediate the transcription of proinflammatory and immunoregulatory genes. Inhibition of BET proteins reduces the expression of important pro-inflammatory factors (IL-22, IL-17, and RORC) and could represent a potential new treatment for psoriasis [184].

#### 4.2.3. Non-Coding RNA

Non-coding RNAs (ncRNAs) are not translated into proteins; instead, they interact with RNA, DNA, and proteins to change their structural properties and affect gene expression (Figure 4) [185]. Non-coding RNAs can be classified as short ncRNAs, intermediate ncRNAs, and long ncRNAs (lncRNAs), depending on their size. Indeed, short ncRNA can be divided into short interfering RNAs (siRNAs), Piwi-interacting RNAs (piRNAs), and microRNAs (miRNAs).

Micro-RNAs (mi-RNAs) are a class of small, single-stranded, non-coding molecules approximately 19–25 base pairs in length and are essential for controlling post-transcriptional gene expression in eukaryotes. Relative preservation of these RNA molecules across time attests to their biological importance. They have been linked to multiple cellular processes such as morphogenesis, proliferation, differentiation, regulation of cellular metabolism, signal transduction, and apoptosis [151,161,186,187,188].

MicroRNAs bind to the targeted messenger RNA (mRNA) strand, resulting in mRNA instability. Unstable mRNA is degraded in the cytoplasm; thus, translation of the corresponding gene is stopped, and the function of the protein is inhibited [186]. Mi-RNAs can also alter DNA expression by modulating DNA methylation, targeting enzymes necessary for DNA methylation, or through histone modifications [185,189]. This epigenetic mechanism is thought to control 1–3% of the human genome, corresponding to at least 30% of the protein-coding genes. Different miRNA species may regulate the same gene, while the expression of different genes may be influenced by the same class of miRNA [186].

##### Non-Coding RNA in AD

Multiple alterations in the expression of specific miRNAs have been observed in the skin and serum of AD patients. They control the expression of genes that mediate Th2 polarization, the activity of Treg, multiple inflammatory processes, the formation of tight junctions, the proliferation and death of epidermal keratinocytes, and the synthesis of cytokines and chemokines [188,190,191,192,193]. Increased levels of 10 miRNA species and decreased levels of 34 have been found in AD skin lesions [191]. Major miRNA alterations related to AD pathogenesis are summarized in Table 3.

The activation of TLR receptors by bacterial lipopolysaccharides and the NFkB-associated activation of the relevant intracellular pathways are two examples of IFN-inducing activities suppressed by miR-146a. Additionally, the *TRAF6*, *IRAK1*, *IRAK2,* and *RELB* genes whose transcripts control the activation of this pathway, are directly inhibited by miR-146. As previously mentioned, NFkB is a transcription factor that mediates the expression of several genes coding proinflammatory cytokines (IL1B, TNFA, CARD10, IRAK1, CCL5, and CXCL). If the expression of these genes is inhibited, there is a reduction in the effectiveness of the innate immune response, which is typically seen in AD [186,190,192].

T helper type 17 (Th17) and Treg cell differentiation depend on miR-155, one of the most upregulated miRNAs in AD patients [191]. In addition, miR-155 blocks the expression of CTLA-4 (cytotoxic T-lymphocyte-associated protein 4), an immune checkpoint receptor that inhibits T-cell responses. Therefore, miR-155 increased expression in the AD skin causes a loss of Treg-dependent regulatory mechanisms, leading to increased effector T-cell proliferation and sustained inflammation [150,191,195,197,198]. Additionally, overexpression of miR-155 in AD is positively correlated with AD severity [199]. A positive correlation between miR-155 expression and Th17 cell percentage (which is increased in AD patients) has also been demonstrated [199].

The miRNA molecules have also been related to the Th2 bias. The Let-7 a-d family acts as an inhibitor of IL-13 and CCR7 synthesis. A decreased expression of Let-7 a-d has been observed in atopic skin, promoting the overproduction of IL-13 and contributing to the Th2 bias characteristic of AD [192]. Furthermore, miR-375 induces TSLP production and promotes Th2 responses in the skin by blocking KLF5 expression [186]. Previous reports have described down-regulation of miR-375 in AD [58,196], but Simpson et al. and Behesti et al. observed a positive correlation between miR-375 levels in breast milk and the risk of AD development in newborn infants [187,200].

Elevated serum levels of mir-151a have been found in AD patients; mir-151a inhibits the expression of the IL12RB2-subunit of the IL-12 receptor in human T helper cells, leading to an increase of the Th2 cells [194,201].

Finally, loss of hsa-miR-26a-5a is linked to increased expression of the enzyme hyaluronan 3 synthase (HAS3) in AD skin; HAS3 is involved in the synthesis of hyaluronic acid, a crucial component of the extracellular matrix [202].

The miRNA molecules have also been studied as possible therapeutic targets. The chemokine CCL22 influences the recruitment of Th2 lymphocytes expressing cutaneous lymphocyte antigen (CLA) into the skin. Dietary treatment with *Salmonella typhimurium* expressing CCL22-targeting miRNA prevented the expression of CCL22 in vivo, resulting in differences in the severity of AD lesions, pruritus, serum levels of IL-4, and serum and skin tissue levels of CCL22 [203].

##### Non-Coding RNA in Psoriasis

Short non-coding RNAs have been strongly associated with the pathogenesis of psoriasis, especially micro-RNAs, which are able to interfere in the immune response and modify skin inflammation as well as keratinocyte proliferation and differentiation [204,205,206]. They are detailed in Table 4.

One of the most downregulated micro-RNAs in psoriasis lesions is miRNA125b, which has been related to inhibition of keratinocyte proliferation and promotion of keratinocyte differentiation [204,205].

Lipocalin 2 (LCN2) has been implicated in the inflammatory process, playing an important role in cell survival and apoptosis. *LCN2* is targeted by miR-383, reducing the expression of JAK3/STAT3 and leading to immune suppression. However, psoriasis lesions have shown downregulation of miR-383, therefore a proinflammatory status [211]. Another micro-RNA downregulated in psoriasis is miR-214-3p, which induces keratinocyte proliferation [212]. Lastly, miR-125a-5p is also downregulated in psoriasis lesions; the methyltransferase EZH2 catalyzes the methylation of histone H3 at lysine 27, repressing transcription of miR-125a-5pn [231] and eventually leading to increased expression of IL-17A-induced cytokines and cell proliferation [213].

Other micro-RNAs are upregulated in psoriasis. MiR-203 is overexpressed in psoriasis lesions, inducing keratinocyte proliferation via inhibition of LXR-α or PPAR-γ [232]. Overexpression of miR-378a (also upregulated in psoriasis) would be induced by IL-17A via the NF-kB pathway [207]. MiR-31 is activated by NF-kB to promote keratinocyte hyperproliferation in psoriatic lesions [208]. High levels of miR-210 have been found in psoriasis lesions and animal models; it stimulates Th17 and Th1 cell differentiation but inhibits Th2 differentiation by suppressing STAT6 and LYN expression [209]. Other micro-RNAs upregulated in psoriasis are miR-200c and miR-155. These micro-RNAs might become potential targets for psoriasis [210,233].

Long non-coding RNAs (>200 nucleotides) have also been associated with psoriasis pathogenesis. LncRNAs modulate gene expression by interacting with transcription factors and proteins that alter the chromatin structure or by acting as competitive endogenous RNAs that limit the accessibility of miRNAs to inhibit their target genes [185,189]. LncRNA-RP6-65G23.1 is significantly upregulated and promotes keratinocyte proliferation and suppression of apoptosis by altering the expression of Bcl-xl, Bcl2, and the p-ERK1/2 p-AKT signaling pathway [214]. MIR31HG is another lncRNA upregulated in psoriasis lesions; it seems to be activated by NF-kB and would participate in keratinocyte proliferation [215]. Several lncRNAs are upregulated in psoriasis, such as MSX2P1 [216], XIST [217], FABP5P3 [218], KLDHC7B-DT [219], or SPRR2C [220], whereas MEG3 [221], GAS5 [222], PRINS [223] or NEAT1 [224] are downregulated in psoriasis.

Finally, circular RNAs (circRNAs) are intermediate single-stranded ncRNAs that show continuous circular structure and covalent coupling between the 5′ and 3′ ends, which makes them unique [189]. CircRNAs participate in the modulation of gene expression or translation of regulatory proteins through their involvement in the degeneration of microRNAs and RNA-binding proteins [234]. Some circRNAs are upregulated in psoriasis, such as circOAS3 [225], circEIF5 [226], circ_0060531 [227], hsa_circ_0003738 [228], or hsa_circ_0061012 [229], whereas circRAB3B is downregulated [230]. The latter acts as an inductor of *PTEN*, suppressing keratinocyte hyperproliferation. CircRAB3B levels in psoriasis lesions are diminished compared to healthy skin.

### 4.3. Proteomics

Proteomics encompasses the examination of protein structure, function, interactions, and post-translational modifications at various phases of the complete set of proteins produced by organisms (proteome). It is widely acknowledged that the regulation of protein expression can operate independently of gene expression; thus, proteomics studies are essential in elucidating the underlying mechanisms responsible for the pathogenesis of diseases [13]. Moreover, proteomic studies allow the identification and characterization of proteins implicated in disease mechanisms without any previous knowledge about these proteins within the disease paradigm [235].

Powerful methods of proteomics allow for extensive examination of proteins potentially implicated in disease pathogenesis [13].

#### 4.3.1. Proteomics in AD

Initial studies of proteomics in AD identified a limited number of proteins potentially involved in AD, such as ALDH1, NCC27, S100/A11, cyclinA2, caldesmon 1 isoform 5, nucleophosmin 1, esterase D, chloride intracellular channel 4, vinculin, and filamin A proteins [236,237,238,239,240,241,242]. S100/A11 induces human beta-defensin (HBD)-3 and FLG expression, and it was found to be significantly downregulated in the presence of IL-4 and IL-13 [239].

Proteomic research with mass spectrometry on tape-strips derived from AD skin has led to the identification of several proteins with pathogenetic implications [243,244,245,246]. Broccardo et al. demonstrated that e-fabp and alpha-enolase were distinctively expressed in AD samples, with increased expression of triosephosphate isomerase and serpin B3, and downregulation of keratin type I cytoskeletal 10, keratin type II cytoskeletal 1, caspase 14 precursor, and dermcidin precursor in comparison to control samples [244]. Additional tape-strip studies found decreased levels of proteins involved in natural moisturizing factor and barrier function compared to non-lesional AD skin; dry skin would predispose to skin infections [243]. Other tape-strip studies have identified 252 and 35 differentially expressed proteins in AD lesional versus healthy skin, respectively [246,247].

Broad proteomic studies with OLINK and SOMAscan have identified dysregulation of several crucial immune and skin barrier AD markers, providing further insight into AD pathogenesis. Proteins associated with general inflammation (MMP12), dendritic cells (CD40, TNF), macrophages (CD163), T-cell activation/migration (IL-2RA, CCL19), Th2 (IL-1RL1, IL-13), Th1 (CXCL9/10/11), Th17/Th22 (IL-12B, IL-17RA), and innate immunity (IL-6, IL-8) are upregulated, whereas proteins related to barrier function are downregulated (FLG, LOR) [248,249,250,251,252,253,254,255]. Although most of these proteins had already been identified by transcriptomic studies, proteomics allowed validation; hence, they could be used as disease severity indicators and even as therapeutic targets [256,257].

Proteomic studies in the skin and/or blood have also been performed in the pediatric AD population [257]. Brunner et al. demonstrated proteomic signs of systemic inflammation in pediatric patients with a 6-month course of moderate-to-severe AD, such as Selectin E (SELE), MMP3, MMP9, MMP10, and urokinase receptor [249]. Cardiovascular risk mediators seen in adult patients were also found in early-onset AD children, with some differences [248,249]: adults with AD exhibited increased expression of a wider array of Th2-associated mediators such as TARC/CCL17, CCL13, and IL-13, as well as the Th17 lymphokine CCL20. Additionally, Th1 markers (CXCL9, CXCL10, and IFN-γ) were upregulated in adults and absent in blood from pediatric AD patients. Therefore, Th1 activation is a marker of AD chronicity and a suitable target to prevent the progression from pediatric AD to chronic AD [249,258].

AD proteomics have been compared to those of other inflammatory diseases. Brunner et al. compared the proteomic blood signatures of moderate-to-severe AD and psoriasis patients. Ten proteins were increased in both diseases compared to controls, including markers of Th1 (IFN-γ, CXCL9, TNF-β/lymphotoxin) and Th17 (CCL20, IL-17C) immune responses, IL-16, BLMH, IL-20, CDCP1, and IL2RA. However, significant differences in their proteomic blood profiles among the two diseases were observed. Markers of systemic inflammation are stronger in AD compared to psoriasis [248,249,257]. Concretely, levels of Th1 (CXCL10, CXCL11), Th2 (IL-13, CCL13, CCL17/TARC, CCL11, IL-10), Th17/Th22 (S100A12) and Th1/Th17/Th22 (IL-12/IL-23p40) associated proteins, and proteins involved in the development and activation of T-cells (CD40L, IL-7, CCL25, IL2RB, IL15RA, CD6, RANKL, TNFRSF9/CD137) and dendritic cells (CD40, FLT3 ligand), mediators associated with atherosclerosis (CX3CL1/fractalkine, CCL8, M-CSF, CXCL5, CCL4, and HGF), factors involved in tissue remodeling (MMP-12, MMP-1, and MMP-10), angiogenesis (VEGF-A), TGF-α, NF-κB activator IKBKG/NEMO, molecules facilitating the chemotaxis of T-cells, B-cells, eosinophils (CCL28), and neutrophils (CXCL5, CXCL6), as well as the soluble cytokine receptors IL-10RB and IL-18R1, were observed to be higher in individuals with AD, compared to psoriasis [248].

Finally, proteomics studies have been used to evaluate the responses of AD to treatment. A tape-strip proteomic study of 353 inflammatory proteins showed a significant decrease of 136 after dupilumab treatment, which suppressed the Th2 pathway without significant modulation of Th1 [252].

#### 4.3.2. Proteomics in Psoriasis

Proteomics studies have provided enhanced insight into the pathogenesis of psoriasis, since pronounced alterations in gene expression may not necessarily lead to substantial modifications at the protein level; conversely, marked shifts in protein expression may not reflect significant alterations in the transcriptome [235]. One of the initial proteomics studies included skin samples taken from individuals with acute guttate psoriasis, chronic plaque psoriasis, and nickel-induced contact eczema; the proteome profile of acute guttate psoriasis closely resembled that of contact eczema, suggesting that the length of disease progression might influence the proteomic signature [259]. Another study showed that cytoskeletal proteins, actin-binding proteins/peptides, and calcium-binding components would participate in the altered protease activity seen in psoriasis [260].

Comparison of psoriasis and healthy skin proteome patterns has led to the identification of several proteins with different expressions in psoriasis [261]. Some of these proteins (SFN protein, peroxiredoxin 2, among others) are involved in cell proliferation, regulatory/balancing system, and inflammatory response. Schonthaler et al. demonstrated higher expression of S100A8, S100A9, and complement C3 in psoriatic epidermis compared to healthy skin [262]. Furthermore, the deletion of the gene coding for S100A9 led to an attenuation of psoriasis-like disease in a murine model [262]. Swindell et al. identified 748 proteins differentially expressed in psoriasis as compared to healthy skin [263]; some of them were categorized as psoriasis-specific, which showed a disproportionate induction by IL-17A in cultured keratinocytes compared to increased but non-specific proteins. Recent evidence has brought to light the importance of peptidase inhibitor 3 (PI3), which is highly expressed in keratinocytes from psoriatic lesions. PI3 is significantly correlated with local gene expression and with disease severity. Hence, PI3 may be a psoriasis-specific biomarker for disease severity and hyper-keratinization [264].

In 2016, Bottoni et al. observed structural protein alterations in the saliva of patients affected by psoriasis, which were similar to those observed in diabetic patients; in both cases, they differed from normal saliva [265].

Utilizing antibody-based methodologies and protein array technology, elevated concentrations of chemoattractants for neutrophils, Th1 cells, monocytes, and dendritic cells were detected in the stratum corneum of individuals with psoriasis; on the other hand, no upregulation of cytokines associated with Th2 cells, which are not implicated in the pathophysiology of psoriasis, was identified [266].

Decreased levels of proteins related to vitamin D regulation and lipid metabolism were observed by Gegotek et al., but their precise role in psoriasis pathogenesis remains unclear [267].

In 2018, Chularojanamontri et al. attempted to define normal skin autoantigens by immunoproteomics in sera from patients with psoriasis and healthy volunteers [268]. By using a healthy skin sample, no significant differences among psoriatic and healthy sera were observed in the patterns of IgG- and IgM-immunoreactivity to both epidermal and dermal autoantigens. These findings suggest that the autoantigens originating from the epidermis and dermis in psoriatic skin, along with the associated humoral immune response, are probably a consequence of downstream processes. In contrast, those linked to upstream mechanisms may likely result from cell-mediated immune mechanisms [268].

Regarding systemic inflammation in patients affected by psoriasis, Brunnet et al. [248] observed significant increases in Th17-associated proteins, proteins involved in coagulation and angiogenesis (t-PA), endothelial activation (IL-6), lipid metabolism (LDL-R), cardiovascular disease risk proteins (galectin-3), mediators of cell adhesion, and T-cell activation (TREML2), among others.

The discovery of biomarkers is one of the major goals of proteomics in psoriasis. A study by Cowen et al. found differences in serum proteome patterns between patients with psoriasis, tumor-staged mycosis fungoides, and healthy participants [269]. Likewise, Williamson et al. identified increased levels of profilin 1, a potential biomarker for psoriasis, in plasma from psoriasis patients [270]. Matsuura et al. identified consistently increased fibrinogen α chain-derived and filaggrin-derived peptide levels in patients with psoriasis vulgaris and psoriatic arthritis [271]. Another protein that holds promise as a potential biomarker is kynureninase, with elevated levels in sera from individuals with psoriasis but no alterations in patients with AD or contact dermatitis [250]. Méhul et al. demonstrated distinct lesional proteomic profiles between psoriasis and cutaneous T-cell lymphoma (CTCL) [272]: psoriasis lesions exhibited differential expression of IL36G and IL37 in contrast to healthy skin, whereas CTCL lesions did not; conversely, CCL27 was differentially expressed in CTCL when compared to healthy skin, with no variations observed in psoriasis lesions. Lastly, desmoplakin, complement C3, polymeric immunoglobulin receptor, and cytokeratin 17 demonstrated significant associations with the Psoriasis Area and Severity Index (PASI) score, indicating their potential as novel biomarkers for disease severity assessment [273].

Proteomics have also been studied regarding the response of psoriasis to treatments. Kolbinger et al. examined the proteomic profile of psoriasis skin and sera before and after secukinumab (anti-IL-17A monoclonal antibody) treatment [274]. Among several proteins examined, β-defensin 2 was found to be correlated with PASI and IL-17 levels. Following the administration of a single dose (300 mg) of secukinumab, the aberrantly regulated antimicrobial peptides, proinflammatory cytokines, and neutrophil chemoattractants reverted to their normal levels. Additionally, a noteworthy decrease in β-defensin 2 was noted. Hence, β-defensin 2 holds potential as a biomarker for IL-17A-mediated psoriatic pathology and as a predictor of response to anti-IL-17A treatment [274].

Proteomics have also been used to identify hepatic and renal toxicity due to systemic therapies for psoriasis. Multiple proteins were identified in urine samples as possible biomarkers of methotrexate-induced hepatic toxicity [275]; however, the reliability of these proteins was not validated with liver biopsy or other alternative methods, such as Fibroscan^®^, magnetic resonance elastography, or FibroTest^®^. Cyclosporine-induced proteomic alterations in human kidney cell lines, which were partially resolved with the administration of *N*-acetylcysteine, remain to be elucidated in psoriatic patients [276].

## 5. Conclusions

This narrative review illustrates the complex nature of AD and psoriasis genetics, epigenetics, and proteomics, involving a wide array of genes and proteins contributing to innate and adaptative immune systems as well as to skin barrier integrity. Rapid advancements in genetic research technology have not only deepened our understanding of the pathophysiology of psoriasis but have also led to the development of new diagnostic biomarkers and novel medications. Psoriasis and AD represent complex diseases with distinct phenotypes; most studies in psoriasis have been focused on plaque psoriasis, and further research is required addressing topographic or morphological variants, such as pustular psoriasis.

## 6. Future Directions

Multi-omics has furthered our knowledge of the pathogenesis of multiple IMIDs and led to the identification of novel disease biomarkers and suitable treatment targets. The goal of personalized medicine, namely to choose treatments taking into account genetic factors and proteomic profiles, seems closer. The molecular underpinnings of psoriasis and AD can be analyzed using a systems biology approach based on microarray data analysis. Various bioinformatics tools can be used to highlight differentially expressed genes, enrich gene ontology, and conduct network analysis to identify critical regulatory pathways. Ultimately, these studies may provide novel therapeutic targets, opening new horizons in medical treatment [277]. Furthermore, response to treatment can be predicted after the first injection of a monoclonal antibody like secukinumab [274]. Multiomic studies may further our understanding of cellular and transcriptomic alterations associated with optimal response to treatments, remission, and potential modification of disease course in psoriasis and other diseases [278,279]. Recently available tools such as mind.px (https://minderahealth.com/psoriasis/) (accessed on 27 December 2023) might contribute to predicting the response of psoriasis to biological treatments and potentially improve the efficiency of treatment.

## Figures and Tables

**Figure 1 ijms-25-01042-f001:**
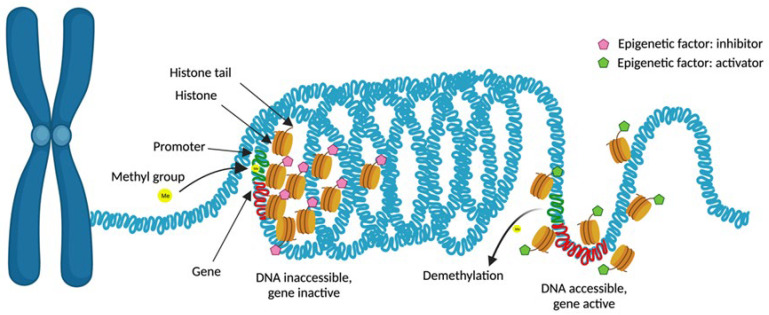
Epigenetic mechanisms at the transcriptional level. Condensed chromatin (heterochromatin) limits gene transcription, whereas open chromatin (euchromatin) allows transcription factor activation and DNA transcription. Methylation of CpG islands in the promoter region of a gene inhibits its expression. Regarding histone modifications, multiple mechanisms can induce modifications in chromatin. Some of them will result in relaxed chromatin, allowing DNA transcription (activating modifications), while others will induce chromatin compaction, hence limiting gene transcription. Created with Biorender.com (accessed on 16 December 2023).

**Figure 2 ijms-25-01042-f002:**
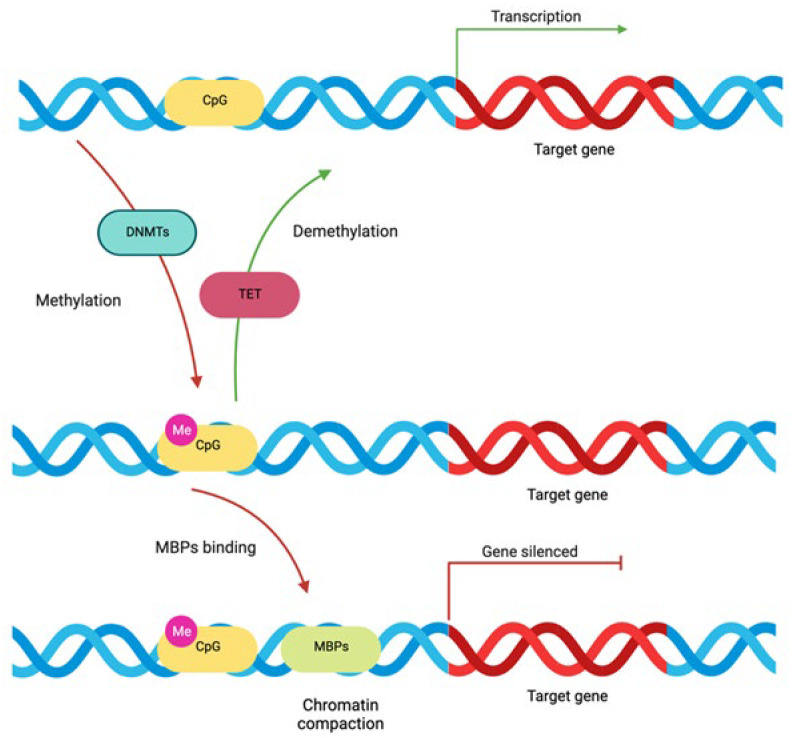
DNA methylation process: Approximately 70% of the proximal promoter regions of the human genome are made up of CpG islands, where cytosine is followed by guanine. Methyltransferases (DNMTs) catalyze the addition of a methyl group to the C5 position of CpG dinucleotides (cytosine becomes 5-methylcytosine), allowing methylcytosine-binding proteins (MBPs) to attach and induce chromatin compaction. Therefore, DNA methylation leads to suppression of gene expression. Conversely, loss of DNA methylation by ten-eleven translocation (TET) enzymes induces chromatin decompaction and gene expression. Created with Biorender.com (accessed on 16 December 2023).

**Figure 3 ijms-25-01042-f003:**
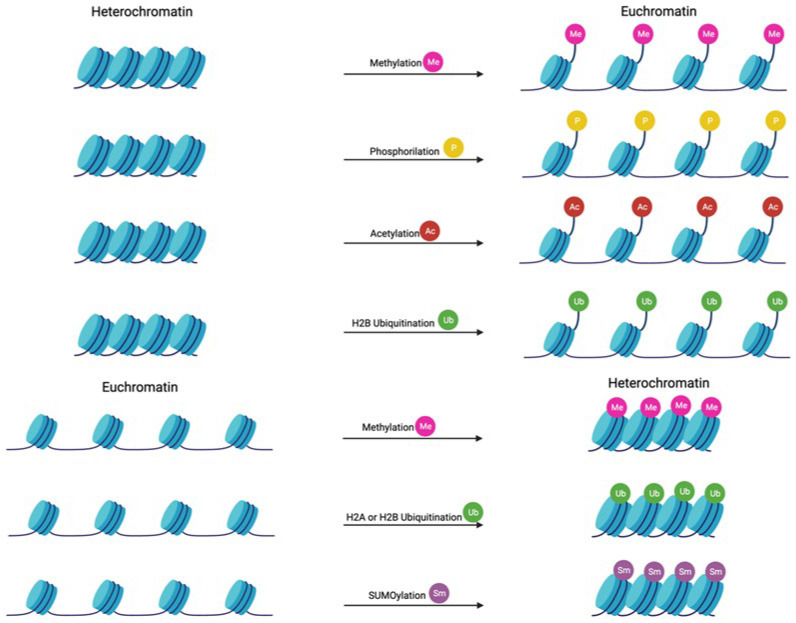
Mechanisms involved in histone modification Nucleosomes are the basic repeating units that package DNA into the nucleus. DNA is wrapped around octamers of histone proteins (two H2A, two H2AB, two H3, and two H4), stabilized by an H1 histone. Histone acetylation and phosphorylation result in chromatin relaxation, allowing DNA transcription. On the other hand, histone SUMOylation and H2A ubiquitination cause chromatin compaction. Histone methylation and H2B ubiquitination may induce both relaxation or compaction of chromatin. Created with Biorender.com (accessed on 16 December 2023).

**Figure 4 ijms-25-01042-f004:**
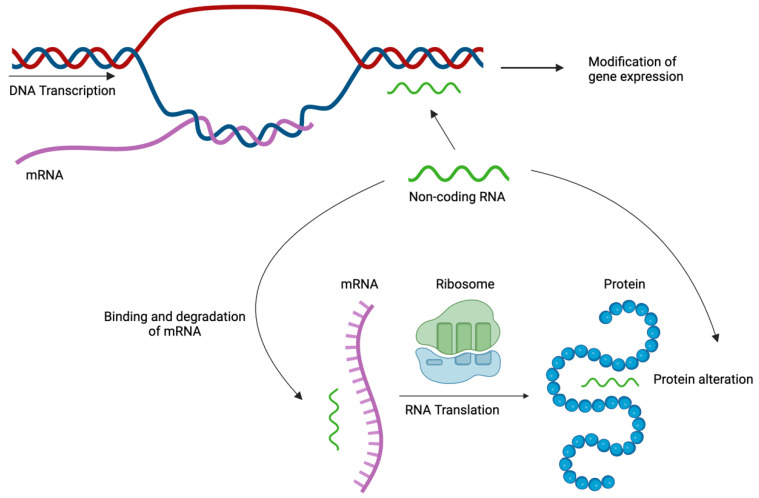
Epigenetic mechanisms at translational level: Non-coding RNAs interact with DNA, mRNA or proteins altering their structure and affecting gene expression, mRNA translation, and protein function. Created with Biorender.com (accessed on 16 December 2023).

**Table 1 ijms-25-01042-t001:** Genomics of AD.

Pathway	Gene	Function
Skin barrier function	FLG [59]	Expression of filaggrin
	OLOV1 [60]	Regulates FLG expression
	MMP9 [61]	Degradation of COL5A3, leading to eczema
	COL29A1 [62]	Lack of expression impacts on epidermis integrity
	SPINK5 [63]	Defective skin cornification
	KIF3A [64]	Increased TEWL
	LCE1D [65]	Impairs epidermis differentiation
	SPRR3 [65]	Impairs structural role
Innate immunity	S100A3 [65]	Chemotactic agent
Adaptative immunity	IL-13 [57]	Mediates Th2 responses
	IL-4 [66]	Inflammation
	IL-10 [67]	Anti-inflammatory (suppressed)
	IL-6 [68]	Stimulates IL-4 and IL-5 production
	IL-6R [69]	Mediates IL-6
	STAT [57]	Inflammation
	TLSP [57]	Regulation of Th2 response
	IRF2 [57]	Th2 polarization
	TLR2 [57]	Suppression of IL-10 by IL-4
	Fcε Rl [57]	Uptake of IgE-associated allergens
Increased AD risk with unknown mechanism	VDR [57]	Unknown
	ACTL9 [64]	Unknown

**Table 2 ijms-25-01042-t002:** Genomics in psoriasis.

Pathway	Gene	Function
Skin barrier function	DEFB4 [92,93]	Secretion of β-defensins
	LCE3B/C [94]	Epidermis differentiation and hyperproliferation
	GJB2 [92,93]	Connexin 26, epidermal gap junction
Innate immunity	C-REL [95]	NF-kB pathway activation
	TRAF3IP2 [96]	NF-kB pathway activation
	CARD14 [87]	NF-kB pathway activation
	MICA [83]	NK, NKT, and T-cells activation
	TNFAIP3 [89]	NF-kB pathway downregulation
	TNIP1 [97]	NF-kB pathway downregulation
	NFKBIA [92]	NF-kB pathway downregulation
	DDX58 [89]	INF pathway and antiviral response
	IFIHI [90]	INF pathway and antiviral response
Antigen presentation	HLA-C*0602 [98]	Antigen presentation
	ERAP1 [99]	Modification of MHC-I-binding peptides
Th1 signaling	IL12B [100]	p40 subunit of IL12
	TYK2 [101]	Downstream molecule of IL12 receptor
	ZC3H12C [89]	Macrophage activation
	STAT5A/B [102]	Signaling pathway of IL2 family cytokines
	ILF3 [102]	IL-2 expression in T-cells
Th17 signaling	TYK2 [103,104]	Downstream molecule of IL23 receptor
	JAK2 [103,104]	Downstream molecule of IL23 receptor
	STAT3 [103,104]	Downstream molecule of IL23 receptor
	SOCS1 [102]	Th17 differentiation
	ETS1 [102]	Th17 differentiation
	IL17RD [105]	IL17 receptor
	IL22 [103]	Differentiation and proliferation of keratinocytes
	TRAF3IP2 [96]	Signaling pathway of IL17A/F
	KLF4 [105]	Regulation of IL17A production

**Table 3 ijms-25-01042-t003:** Epigenomics in AD.

miRNAs	Mechanism of Action	Consequence	Target Cells	Target mRNA
miR-10a-5p [58] (up-regulated)	Inhibits keratinocytes proliferation	Impaired skin barrier function	Epidermal keratinocytes	HAS3
miR-29b [194] (up-regulated)	Promotion of INF-γ-induced keratinocyte apoptosis	Epithelial barrier dysfunction	Epidermal keratinocytes	BCL2L2
miR-124 [194] (down-regulated)	Inhibition of inflammatory responses	Pro-inflammatory status	Epidermal keratinocytes	RELA (p65 subunit of NF-κB)
miR-143 [194] (down-regulated)	Suppression of IL-13-induced dysregulation of skin barrier proteins	Down-regulation of filaggrin, loricrin, and involucrin due to IL-13	Epidermal keratinocytes	IL-13Rα1
miR-146a [58] (down-regulated)	Suppresses the expression of many pro-inflammatory factors	Stronger inflammatory reaction, increased inflammatory cells in the dermis and elevation of inflammatory factors	Epidermal keratinocytes	IL1B, TNFA, CARD10, IRAK1, CCL5 and CXCL
miR-151a [58,195] (up-regulated)	Inhibition of IL-12 signaling	Promotes Th2 differentiation	T helper cells	IL12RB2
miR-155 [58,195] (up-regulated)	Inhibition of CTLA-4 in T-cells Inhibition of PKIα	Promotion of Th17 differentiation Epidermal thickening and inhibition of tight junction formation	T-cells Epidermal keratinocytes	CTLA-4 PKIα
miR-223 [58] (up-regulated)	Unknown	Lower Treg cell count	T-cells	Unknown
Let-7 a-d [58] (down-regulated)	Inhibitors of IL-13 and CCR7 synthesis	Up-regulation of IL-13 expression, leads to Th2 differentiation	T-cells	IL-13 and CCR7
miR-375 [58] (up-regulated)	Expression of TSLP by blocking KLF5 TSLP suppresses Th1 responses	Enhance Th2 responses	T-cells	KLF5
hsa-miR-26a-5a [58] (down-regulated)	Synthesis of hyaluronic acid by HAS3 inhibition	Reduced synthesis of hyaluronic acid	Keratinocytes	HAS3
miR-21 [196] (up-regulated)	Inhibiting IL-12	Promotion of Th2	T-cells	IL-12
Other miRNA expression changes in atopic skin [58]: ↑ miR-17-5p, ↑ miR142-3p/5p, ↓ miR-122a, ↓ miR-326, ↓ miR-133b, ↓ miR-125b, ↓ miR375, ↓ miR193c, ↓ miR365				

↑: up-regulated. ↓: down-regulated.

**Table 4 ijms-25-01042-t004:** Epigenomics in psoriasis.

Non-Coding RNAs	Molecule	Function
miRNA upregulated	miR-378a [207]	Psoriatic inflammation
	Mir-31 [208]	Keratinocyte proliferation
	mir-210 [209]	Inflammation
	miR-200c [210]	Associated with PASI
	miR-155 [195]	Psoriatic inflammation
	miR-203 [188]	Keratinocyte proliferation
miRNA downregulated	miRNA125b [205]	Keratinocyte proliferation and differentiation
	mir-383 [211]	Keratinocyte apoptosis and inflammation
	214-3p [212]	Cell cycle check-points and keratinocyte proliferation
	miR-125a-5p [213]	Keratinocyte proliferation
lnc upregulated	lncRNA-RP6- 65G23.1 [214]	Immune response, keratinocyte proliferation, apoptosis suppression
	MIR31HG [215]	Keratinocyte proliferation
	MSX2P1 [216]	Keratinocyte proliferation
	XIST [217]	Keratinocyte proliferation
	FABP5P3 [218]	Keratinocyte proliferation and inflammation
	KLDHC7B-DT [219]	Keratinocyte proliferation and inflammation
	SPRR2C [220]	Keratinocyte proliferation and apoptosis
lnc downregulated	MEG3 [221]	Keratinocyte proliferation and apoptosis
	GAS5 [222]	Related to psoriasis severity
	PRINS [223]	Keratinocyte proliferation and inflammation
	NEAT1 [224]	Keratinocyte proliferation
circRNA upregulated	circOAS3 [225]	Keratinocyte proliferation and apoptosis
	circEIF5 [226]	Keratinocyte proliferation
	circ_0060531 [227]	Keratinocyte proliferation, migration and inflammation
	hsa_circ_0003738 [228]	Treg modulation
	hsa_circ_0061012 [229]	Keratinocyte proliferation and migration
circRNA downregulated	circRAB3B [230]	Keratinocyte proliferation

## Data Availability

No new data were created or analyzed in this study. Data sharing is not applicable to this article.

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
