# Peer review of "Multi-Omics Approach to Improved Diagnosis and Treatment of Atopic Dermatitis and Psoriasis"

_ijms, 2024, doi:10.3390/ijms25021042_

Round 1
Reviewer 1 Report
Comments and Suggestions for Authors
The review Multiomics approach to improved diagnosis and treatment of atopic dermatitis and psoriasis may be published in the IJMS journal after minor revision.
Comments on the article.
What is unique about this review compared to previously published ones (see https://doi.org/10.3390/ijms23105518, https://doi.org/10.3390/ijms23094898)? In addition, the mechanism of development of psoriasis has been previously described (see works 10.7759/cureus.47006, 10.3390/ijms20061475).
It is necessary to provide a diagram of the pathogenesis of psoriasis in the form of a picture for better comprehension of the text. You should also consider the pathomechanisms of various subtypes of psoriasis – guttate psoriasis, pustular psoriasis, nail psoriasis, etc.
Provide links to publications in all review tables.
The S100A8 and S100A9 genes are among the leading genes differentially expressed in human psoriatic skin. A more detailed description of the defective activation of expression of these genes is needed.
Author Response
Reviewer 1:
The review Multiomics approach to improved diagnosis and treatment of atopic dermatitis and psoriasis may be published in the IJMS journal after minor revision.
Comments on the article.
What is unique about this review compared to previously published ones (see https://doi.org/10.3390/ijms23105518, https://doi.org/10.3390/ijms23094898)? In addition, the mechanism of development of psoriasis has been previously described (see works 10.7759/cureus.47006, 10.3390/ijms20061475).
We sincerely appreciate your thoughtful review.
The comprehensive analyses provided in the reviews: https://doi.org/10.3390/ijms23105518 and https://doi.org/10.3390/ijms23094898, extensively explore the genomics and pathogenesis of psoriasis and atopic dermatitis. Nevertheless, it is worth noting that these reviews do not encompass information pertaining to epigenomics and proteomics.
While the mechanisms underlying psoriasis development have been well-documented, it is noteworthy that both reviews overlook aspects such as the proteomic profile and recent findings. For instance, the identification of a proteomic profile indicative of a favorable response to secukinumab represents a significant advancement in recent research. This newly discovered evidence adds a noteworthy dimension to the existing body of knowledge.
It is necessary to provide a diagram of the pathogenesis of psoriasis in the form of a picture for better comprehension of the text. You should also consider the pathomechanisms of various subtypes of psoriasis – guttate psoriasis, pustular psoriasis, nail psoriasis, etc.
We appreciate your comments. While the diagram suggested by the reviewer would undoubtedly be of interest, we consider that the pathogenesis of psoriasis has been widely described previously and it is not the focus of our paper.
We have added information about the pathomechanisms of psoriasis subtypes (page 3, line 121):
Psoriasis encompasses multiple clinical variants such as, pustular psoriasis, guttate psoriasis, and nail psoriasis, among others. Guttate psoriasis was previously thought to be closer to contact dermatitis than psoriasis, however novel insights by using gene ex-pression profiling and gene set enrichment scores has been observed that is more similar to chronic psoriasis [48]. Regarding pustular psoriasis, in contrast to psoriasis vulgaris, where the IL-23/17 axis plays a pivotal role, pustular psoriasis shows hyperactivation of innate immunity, prominently involving the IL-36 axis [49]. Analysis of gene expression in skin biopsy specimens obtained from individuals affected with either generalized pustular psoriasis (GPP) or plaque psoriasis has unveiled discernible patterns. Specifically, GPP lesions manifest increased expression of IL-1 and IL-36, coupled with diminished levels of IL-17A and interferon-c, when juxtaposed with lesions characteristic of plaque psoriasis [47]. The pathomechanisms of ungual psoriasis remain elusive; however, a variant in IL1RN has been identified in patients with nail psoriasis. IL1RN functions to regulate the proinflammatory activity of IL-1A. The latter has been demonstrated to induce nail changes, suggesting a potential association with nail involvement in patients affected by psoriasis [48,49].
Provide links to publications in all review tables.
References have been provided.
The S100A8 and S100A9 genes are among the leading genes differentially expressed in human psoriatic skin. A more detailed description of the defective activation of expression of these genes is needed.
Thank you for your suggestions. We have added information in page 3 line 110: S100A8/A9 are overexpressed in keratinocytes and innate immune cells, and their transcripts are significantly overexpressed in psoriasis lesions compared to non-lesional psoriasis or atopic dermatitis (AD) skin [45]. Furthermore, psoriasis treatment has been shown to reduce S100A8/A9 levels. Christmann et al. identified an induction of S100-alarmins in an imiquimod-induced murine model of psoriasis-like skin inflammation, which was associated with increased expression of IL-1α, IL-6, IL-17A, or TNF [46]. However, recent evidence has observed that lower epidermal levels of S100A9 in mice lead to more severe psoriasis skin lesions [47].
Reviewer 2 Report
Comments and Suggestions for Authors
This review focuses on the current knowledge in genomics, epigenomics, and proteomics of atopic dermatitis and psoriasis. A literature search was conducted for articles published until November 30, 2023. The idea of the work is novel in manuscript. However, I still have some questions about some parts in the literature.
1. Currently, hot topics in multiomics research, such as cuproptosis, Ferroptosis, and pyroptosis, are not discussed in this article. This article only discusses the insufficiency of methylation.
2. This article lacks mechanism diagrams. A specific mechanism diagram should be added to each chapter.
3. Since this is a multiomics review, it should cover genomics, transcriptomics, single-cell transcriptomics, proteomics, and metabolomics. This article does not provide a detailed discussion following this framework.
Author Response
- Currently, hot topics in multiomics research, such as cuproptosis, Ferroptosis, and pyroptosis, are not discussed in this article. This article only discusses the insufficiency of methylation.
Thank you for your kind review. We have added a new section facing cuproptosis, ferroptosis and pyroptosis in AD and psoriasis (page 9, line 391):
4.1.4. Ferroptosis, pyroptosis and cuproptosis
Cell death plays a critical role in embryonic development, cell fate determination, and mantainance of immune homeostasis. It is categorized into necroptosis, apoptosis, pyroptosis, cuproptosis, ferroptosis, and necrosis. In this chapter we will focus on cuproptosis, ferroptosis, and pyroptosis [127].
Ferroptosis, a form of iron-dependent cell death, results in a toxic accumulation of reactive oxygen species. The dysruption of iron homeostasis and the oxidation of phospholipids can instigate the occurrence of ferroptosis. Due to its distinctive mechanism, ferroptosis may play a role in determining cellular outcomes, the advancement of inflammatory processes, and various pathological conditions [128].
Pyroptosis, a recently identified form of programmed cell death, is orchestrated by pyroptotic caspases [127]. This modality is characterized by the prompt rupture of the plasma membrane, leading to the release of inflammatory intracellular contents. Recognition of pathogen-associated molecular patterns (PAMPs), damage-associated molecular patterns (DAMPs), and lipopolysaccharide (LPS) by specific inflammasomes and caspases, induces the activation of pyroptosis pathways. There are two pyroptosis pathways: canonical and non-canonical. The former is caused by PAMPs and DAMPs, which permit the formation of inflammosomes. Inflammasomes cleave procaspase-1 into caspase-1; activated capsase-1 cleaves gasdermin D (GSDMD) and the maturation of IL-1B and IL-18. Whereas the non-canonical pathway lipopolysaccharide can activate caspase-4/5/11 to induce pyroptosis by cleavage of GSDMD. Increasing evidence suggests the profound involvement of pyroptosis in infectious diseases, hematologic disorders, and tumorigenesis. Furthermore, activated inflammasomes triggered by both PAMPs and DAMPs are implicated in the initiation of chronic inflammation and autoimmune diseases.
Cuproptosis represents a novel type of cellular death associated with mitochondrial metabolism and facilitated through protein lipoylation [129]. The occurrence of cell death induced by copper ionophores depends predominantly upon the intracellular accumulation of copper. Research findings indicate that FDX1 and protein fatty acylation play pivotal roles as regulators in the context of cuproptosis.
4.1.4.1. Ferroptosis, pyroptosis and cuproptosis in AD
Pyroptosis has been related to AD pathogenesis [130]. Research has focused on the identification of pyroptosis related genes (PRGs). Recent studies have identified an increased expression of inflammasome PRGs, such as: NLR Family Pyrin Domain Containing 3 (NLRP3) and Absent In Melanoma-2 (AIM2) in AD, which have been related to epidermal inflammation [131,132]. Also, genes of gasdermin C and GSDMD were found to be highly expressed in lesional skin-derived keratinocytes [133,134]. TNF, which plays a role in the pathogenesis of allergic inflammation in AD, forms a complex with caspase-8 and GSDMC. Pyroptosis mediated by the complex TNF-Caspase8-GSDMC could trigger keratinocyte inflammation and death [134]. All these findings suggest that PRGs may play a pivotal role in AD pathogenesis, and ultimately the identification of pyrptosis related biological markers (PRBMs) could provide a novel perspective on the diagnosis and treatment of AD [135].
To our knowledge, no ferroptosis or cuproptosis alterations related to AD pathogenesis have been identified yet.
4.1.4.2. Ferroptosis, pyroptosis and cuproptosis in psoriasis
Ferroptosis is an iron-dependent, lipid peroxidation-mediated cell death pattern that is intricately linked to inflammation within psoriatic lesions [136]. The extent of ferroptosis impact on psoriasis remains largely undetermined, but keratinocytes isolated from psoriatic lesions exhibit irregularities in lipid metabolism and expression. At the individual cell level, there is pronounced activation of lipid oxidation and peroxidation in psoriasis lesional keratinocytes[137].
Ferroptosis can be activated by the generation of reactive oxygen species (ROS) [136]. ROS induce GPX4 (glutathione peroxidase 4) deletion. In normal conditions, GPX4 inhibits ferroptosis [138]. GPX4 deletion decreases keratinocyte adhesion and increases intracellular lipid peroxidation [139]. GPX4 expression is decreased in psoriatic skin lesions, compared to healthy skin. Furthermore, augmentation in the cellular import of iron indicates activation of ferroptosis in psoriasis lesions [140]. Other genes which have been found to be related to ferroptosis and to regulate the immune microenvironment in psoriasis are: PEBP1, PRKAA2, and ACSF2 [137]. PEBP1 mediates ferroptosis vulnerability due to GPX4 deletion, and PRKAA2 induces ferroptosis by inhibiting the transcription of SLC7A11 [141]. Finally, a positive correlation between lipidoxidation and the Th22/Th17 pathway at a single-cell level has been demonstrated: Fer-1 (a specific inhibitor of ferroptosis) decreases inflammation in mice with imiquimod-induced psoriasis, with significantly decreased expression of TNF, IL-6, IL-1a, IL-1b, IL-17, IL-22, and IL-23 [138].
Pyroptosis plays a crucial role in the pathogenesis of disorders with an aberrant Th17 immune response, like psoriasis [142]. Keratinocytes apparently engage cell death by pyroptosis, paradoxically accelerating cell proliferation in the pathogenic environment of psoriasis. In addition, GSDMD-mediated pyroptosis would contribute to the inflammatory environment by allowing the release of DAMPs and inflammatory cytokines. Effective inhibition of pyroptosis with topical application of disulfiram 2% and 5% has been reported to reduce the psoriasis-like inflammation in a murine model [142].
Cuproptosis and elevated copper levels in serum may be involved in the pathogenesis of psoriasis by mechanisms largely unknown [129]. The accumulation of intracellular copper leads to cell death, and the cuproptosis related genes MTF1, ATP7B, and SLC31A1 are increasedly expressed in patients with psoriasis compared to patients without psoriasis[129].
- This article lacks mechanism diagrams. A specific mechanism diagram should be added to each chapter.
We sincerely appreciate your suggestion. Pathway analyses are documented in various papers; however, it is essential to note that the results are study-specific and cannot be reproduced without obtaining proper permissions (e.g. Mohapatra SS, Mohapatra S, McGill AR, Green R. Molecular mechanism-driven new biomarkers and therapies for atopic dermatitis. J Allergy Clin Immunol. 2020 Jul;146(1):72-73. doi: 10.1016/j.jaci.2020.04.039.)
- Since this is a multiomics review, it should cover genomics, transcriptomics, single-cell transcriptomics, proteomics, and metabolomics. This article does not provide a detailed discussion following this framework.
Due to the considerable length of the manuscript, reaching up to 10,000 words, we found it necessary to constrain the scope of our review, focusing specifically on genomics, epigenomics, and proteomics. The elucidation of this matter is provided in the introductory section:
Due to the considerable length of the manuscript, we found it necessary to constrain the scope of our review. Hereby, we review the current knowledge of genomics, epigenomics, and proteomics in psoriasis and atopic dermatitis. Nevertheless, intriguing facts are also being unveiled by other omics technologies. For example, the combination of proteomics, single cell transcriptomics and spatial transcriptomics by Mitamura et al. uncovered the cellular crosstalk between the immune cells involved in skin lesions of atopic dermatitis [11]. Also, metabolomics has enabled the identification of urine metabolites associated to IMIDs, which could be valuable for their diagnosis [12].
Reviewer 3 Report
Comments and Suggestions for Authors
This narrative review illustrates the complex nature of AD and psoriasis genetics, epigenetics and proteomics, involving a wide array of genes and proteins contributing to innate and adaptative immune systems, as well as to skin barrier integrity.
Rapid advancements in genetic research technology have not only deepened our understanding of the pathophysiology of psoriasis but also led to development of new diagnostic biomarkers and novel medications.
Psoriasis and AD represent complex diseases with distinct phenotypes; most studies in psoriasis have been focused on plaque psoriasis, and further research is required addressing topographic or morphological variants, such as pustular psoriasis.
Multiomics has furthered our knowledge on the pathogenesis of multiple IMIDs and led to the identification of novel disease biomarkers and suitable treatment targets.
The goal of personalized medicine, namely to choose treatments taking into account genetic factors and proteomic profiles, seems closer.
Response to treatment can be predicted after the first injection of a monoclonal antibody like secukinumab.
Multiomic studies may further our understanding of cellular and transcriptomic alterations associated with optimal response to treatments, remission, and potential modification of disease course in psoriasis and other disea

Author Response
This narrative review illustrates the complex nature of AD and psoriasis genetics, epigenetics and proteomics, involving a wide array of genes and proteins contributing to innate and adaptative immune systems, as well as to skin barrier integrity.
Rapid advancements in genetic research technology have not only deepened our understanding of the pathophysiology of psoriasis but also led to development of new diagnostic biomarkers and novel medications.
Psoriasis and AD represent complex diseases with distinct phenotypes; most studies in psoriasis have been focused on plaque psoriasis, and further research is required addressing topographic or morphological variants, such as pustular psoriasis.
Multiomics has furthered our knowledge on the pathogenesis of multiple IMIDs and led to the identification of novel disease biomarkers and suitable treatment targets.
The goal of personalized medicine, namely to choose treatments taking into account genetic factors and proteomic profiles, seems closer. Response to treatment can be predicted after the first injection of a monoclonal antibody like secukinumab. Multiomic studies may further our understanding of cellular and transcriptomic alterations associated with optimal response to treatments, remission, and potential modification of disease course in psoriasis and other diseases.
Thank you for your kind review.
Round 2
Reviewer 2 Report
Comments and Suggestions for Authors
The revised manuscript can be published